# FusionOpt-Net: A Transformer-Based Compressive Sensing Reconstruction Algorithm

**DOI:** 10.3390/s24185976

**Published:** 2024-09-14

**Authors:** Honghao Zhang, Bi Chen, Xianwei Gao, Xiang Yao, Linyu Hou

**Affiliations:** Beijing Electronic Science and Technology Institute, Beijing 100070, China; 20222942@mail.besti.edu.cn (H.Z.); 20223919@mail.besti.edu.cn (B.C.); 20222957@mail.besti.edu.cn (X.Y.); 20223915@mail.besti.edu.cn (L.H.)

**Keywords:** compressive sensing, image reconstruction, FISTA, transformer, deep learning

## Abstract

Compressive sensing (CS) is a notable technique in signal processing, especially in multimedia, as it allows for simultaneous signal acquisition and dimensionality reduction. Recent advancements in deep learning (DL) have led to the creation of deep unfolding architectures, which overcome the inefficiency and subpar quality of traditional CS reconstruction methods. In this paper, we introduce a novel CS image reconstruction algorithm that leverages the strengths of the fast iterative shrinkage-thresholding algorithm (FISTA) and modern Transformer networks. To enhance computational efficiency, we employ a block-based sampling approach in the sampling module. By mapping FISTA’s iterative process onto neural networks in the reconstruction module, we address the hyperparameter challenges of traditional algorithms, thereby improving reconstruction efficiency. Moreover, the robust feature extraction capabilities of Transformer networks significantly enhance image reconstruction quality. Experimental results show that the FusionOpt-Net model surpasses other advanced methods on various public benchmark datasets.

## 1. Introduction

The demand for IoT solutions is currently experiencing significant growth, with the IoT community and the majority of IoT terminal markets showing strong positive sentiment. In terms of device scale, a report by IoTAnalytics projects that there will be approximately 27 billion IoT devices by 2025 [1]. Vision is considered the most critical and convenient form of perception, and visual data (such as images and videos) have become the preferred medium for information exchange within the IoT. It is reported that approximately 1.81 trillion photos are taken globally each year, and by 2030, the total number of photos taken is expected to reach 28.6 trillion. Of these, an estimated 6% will be shared and transmitted over the IoT to meet various needs. However, the rapid annual growth in the volume of data that IoT devices need to process has led to a significant challenge. Many IoT devices are resource-constrained, with limited data processing capabilities and strict energy consumption requirements. This necessitates the development of new solutions to efficiently handle data and support the execution of intelligent tasks within the IoT. For example, Yiting Lin [2] proposed an image compression and reconstruction algorithm based on compressed sensing that addresses these challenges.

Compressed sensing (CS) [3] is a recently developed signal acquisition, processing, and compression technique. It breaks through the limitations of the traditional Nyquist/Shannon [4,5] sampling theorem. Since its introduction by Candes, Tao, and Donoho in 2006, this theory has shown that it is possible to recover high-dimensional sparse signals from a small number of linear, non-adaptive measurements. This is achievable by solving an optimization problem, even when the measurement count is substantially less than what the Nyquist/Shannon theorem prescribes. Despite reducing the sampling rate, CS still allows for the efficient recovery of signals, making it a promising approach for IoT applications.

To address the optimization problem in CS, several efficient algorithms have been developed, including iterative hard thresholding (IHT) [6], iterative shrinkage-thresholding algorithm (ISTA) [7], fast iterative shrinkage-thresholding algorithm (FISTA) [8], and approximate message passing (AMP) [9].

The original CS problem involves finding the sparsest solution, defined as
(1)$$\min_{x}{∥x∥}_{0}\;\mathrm{subject}\;\mathrm{to}\;y=\Phi x.$$

Given noisy measurements y, the CS problem is typically solved as
(2)$$\min_{x}\frac{1}{2}{∥\Phi x-y∥}_{2}^{2}+\lambda {∥x∥}_{1}$$
where ${∥\Phi x-y∥}_{2}^{2}$ represents the data fidelity term, and λ is a regularization parameter. For example, ISTA updates the estimate as
(3)$$r^{\left(k\right)}=x^{\left(k\right)}-\rho \Phi^{T}\left(\Phi x^{\left(k\right)}-y\right)$$
and applies the thresholding operation:(4)$$x^{\left(k+1\right)}=\arg\min_{x}\frac{1}{2}∥x-r^{\left(k\right)}{∥}_{2}^{2}+\lambda {∥x∥}_{1}$$
where *k* is the iteration step, and ρ controls the convergence speed and accuracy of the thresholding process.

The primary drawback of traditional reconstruction algorithms lies in their slow convergence speed. Due to the requirement for extensive iterations, significant computational resources are consumed when dealing with large-scale or high-dimensional datasets, making it difficult to meet efficiency demands. Additionally, the performance of traditional reconstruction algorithms highly depends on the selection of preset parameters, such as regularization thresholds and step sizes. These parameters often need experimental tuning, which increases the algorithm’s complexity and usability challenges.

In recent years, with the significant success of emerging deep learning (DL) techniques in computer vision, numerous DL-based models have been proposed for compressive sensing (CS) image reconstruction, such as LISTA [10], ISTA-Net [11], and FISTA-Net [12]. Compared to traditional algorithms, these DL-based CS algorithms leverage extensive training data to learn complex signal features, thereby achieving higher quality reconstructions that better preserve image details and textures. Moreover, deep learning models autonomously learn features from data without the need for manually designed feature extraction methods, addressing the hyperparameter issues of traditional algorithms like FISTA. This capability of automatic feature learning endows deep learning methods with notable advantages in handling complex and high-dimensional data. In fact, in addition to capturing local image features, the global spatial information of images is crucial. However, relying solely on convolutional neural networks (CNNs) to comprehensively learn global information may be limited due to the inherent constraints of stacked convolutional layers, such as effective receptive fields and the issue of redundant filters from over-parameterization. This approach could potentially constrain image reconstruction performance. Addressing this challenge, Shen et al. proposed the TransCS model [13], which introduces a custom ISTA-based Transformer backbone. This model applies iterative gradient descent updates and soft thresholding operations to represent the global spatial relationships among image patches. Nevertheless, the Transformer architecture in TransCS remains computationally complex. To simplify iteration counts and reduce computational resource consumption, we present FusionOpt-Net, a CS model based on Transformer and FISTA algorithms. FusionOpt-Net incorporates a momentum factor and novel sequences to accelerate convergence, while integrating Transformer’s global features to achieve superior image reconstruction performance.

FusionOpt-Net, with its high image reconstruction performance and fast computational speed, is particularly effective in eliminating blocking artifacts and restoring image details even at low sampling rates. This makes it highly suitable for real-time image reconstruction tasks, such as video compression and transmission [14]. Moreover, FusionOpt-Net demonstrates excellent image reconstruction capabilities in noisy environments, maintaining high PSNR and SSIM metrics even in the presence of multiple levels of Gaussian noise. This suggests that the model is applicable in fields requiring high-quality image reconstruction in noisy conditions, such as medical imaging [15] and remote sensing image processing. The primary contributions of this paper are as follows:We propose an innovative framework that integrates FISTA with Transformer networks. Through this integration, we leverage the fast convergence properties of FISTA and the powerful feature extraction capabilities of Transformer networks to significantly enhance the performance of compressive sensing image reconstruction;We conducted experiments on several public datasets to validate that the proposed FusionOpt-Net model outperforms other image-compression-aware reconstruction models significantly in terms of visual representation and quantitative performance metrics.

## 2. Related Work

### 2.1. FISTA Algorithm

The fast iterative shrinkage-thresholding algorithm (FISTA) is an accelerated gradient-based method designed to solve sparse linear inverse problems. It builds upon the traditional iterative shrinkage-thresholding algorithm (ISTA) by incorporating momentum acceleration, which significantly enhances convergence speed. Due to its efficiency, FISTA has been widely adopted in fields such as compressed sensing and image reconstruction.

FISTA is formulated to solve optimization problems of the form
(5)$$\min_{x}\left\{F(x)=f(x)+g(x)\right\}.$$
Here, f(x) represents a smooth convex function, typically associated with data fidelity, and is expressed as
(6)$$f\left(x\right)=\frac{1}{2}{∥Ax-b∥}_{2}^{2}$$
g(x) is a non-smooth but convex regularization term, often chosen as the L1 norm:(7)$$g\left(x\right)={\lambda ∥x∥}_{1}.$$

**Initialization:** The algorithm starts with an initial point $x_{1}=y_{1}$ and an initial step size parameter $t_{1}=1$.

**Iterative Update:** In each iteration, the following update rules are applied:(8)$$x_{k+1}={\mathrm{prox}}_{\gamma g}\left(y_{k}-\gamma \nabla f\left(y_{k}\right)\right)$$
where γ is the step size, typically set to $\gamma =\frac{1}{L}$, with *L* being the Lipschitz constant of the smooth function f(x). The function ${\mathrm{prox}}_{\gamma g}\left(v\right)$ denotes the proximal operator associated with g(x), defined as
(9)$${\mathrm{prox}}_{\gamma g}\left(v\right)=\arg\min_{x}\left\{\frac{1}{2\gamma }{∥x-v∥}_{2}^{2}+g\left(x\right)\right\}.$$
The acceleration parameter $t_{k+1}$ is then updated as follows:(10)$$t_{k+1}=\frac{1+\sqrt{1+4t_{k}^{2}}}{2}.$$
Finally, the auxiliary variable $y_{k+1}$ is updated using
(11)$$y_{k+1}=x_{k+1}+\frac{t_{k}-1}{t_{k+1}}\left(x_{k+1}-x_{k}\right).$$

**Termination:** The iterative process continues until a predefined convergence criterion is satisfied, such as when the difference $∥x_{k+1}-x_{k}∥$ falls below a certain threshold.

FISTA’s primary advantage lies in its enhanced convergence rate and ease of implementation. Specifically, compared to traditional gradient descent and ISTA, FISTA achieves a faster convergence rate by utilizing Nesterov’s momentum. This improvement leads to a theoretical convergence rate of $O(1/k^{2})$ compared to ISTA’s O(1/k), making it highly effective for large-scale sparse problems. Furthermore, despite the inclusion of momentum, FISTA maintains a computational complexity comparable to ISTA, ensuring both efficient implementation and execution. Moreover, FISTA exhibits great flexibility, as it can be adapted to various regularization terms, such as L1 and L2 norms, making it applicable to a broad range of sparse optimization problems. This adaptability has contributed to FISTA’s widespread use as a reliable tool in areas like compressed sensing and image reconstruction.

### 2.2. Transformer

The Transformer [16] is a deep learning architecture known for its reliance on the self-attention mechanism, which allows it to capture long-range dependencies in sequential data more effectively than traditional RNNs. Its multi-head attention further enhances the model’s ability to learn diverse patterns by processing multiple attention layers in parallel. Unlike RNNs, the Transformer operates with full parallelism, significantly improving training efficiency. Additionally, positional encoding is used to maintain the order of sequences, while residual connections and layer normalization ensure stable training. These features make the Transformer a highly flexible and powerful model, applicable across various domains including natural language processing and computer vision.

While the Transformer has become the standard for NLP tasks, its application in visual tasks still requires more exploration. An experimental approach to image compressed sensing (CS) is CSformer, which adopts a dual-stream, black-box strategy to merge intermediate features from both Transformer and CNN. In contrast, another work, TransCS, applies global attention to natural images through an iterative process, which can be regarded as an unfolded ISTA recovery framework. This method iteratively conducts gradient descent updates and soft-thresholding, providing well-defined interpretability. Additionally, by integrating Transformer and CNN into a hybrid architecture, TransCS excels at managing the relationships between high-level visual semantic features. Consequently, TransCS capitalizes on the strengths of both Transformer and CNN for image CS, learning global dependencies and local features of image patches, leading to hybrid image reconstruction with high recovery quality. However, the traditional ISTA algorithm used in TransCS, although resolving inherent hyperparameter challenges, suffers from slow convergence and low efficiency. To overcome this limitation, we combine the FISTA algorithm with Transformer, incorporating learnable momentum, which not only accelerates convergence but also preserves high reconstruction accuracy.

### 2.3. Deep Compressed Sensing

The fundamental idea behind deep compression sensing (DCS) is to utilize a neural network to learn the complex relationship between measurements and the original signal. This approach enhances both the speed and precision of the reconstruction process, thereby improving the overall performance in image sampling and reconstruction. Typically, DCS aims to minimize the expression $∥x-g_{\Phi }{\left(y\right)∥}_{2}^{2}$, where x represents the source signal, and y denotes the observation, serving as the network input. The inverse transformation function, determined by the network’s parameters Φ, is optimized through this process. With the ongoing advancements in deep learning, a growing number of DCS algorithms are being introduced.

These algorithms generally fall into two main categories. The first type integrates traditional CS algorithms with deep learning, employing neural networks for both implementation and computation in an iterative manner. This method maintains the stability and dependability of conventional algorithms while enhancing reconstruction quality and speed through deep learning. For example, ISTA-Net substitutes the sparsity constraints in the linear transform domain of traditional optimization-based spreading algorithms with constraints in the nonlinear transform domain of the network. A similar approach is employed in ADMM-CSNet [17], which builds upon the ADMM algorithm. Although these models utilize a data-driven method for reconstruction, they continue to rely on the traditional, manually designed sensing matrix within the sampling module, potentially limiting reconstruction performance. Additionally, NeumNet [18] was introduced by Gilton et al. as a solution for image inverse problems using the Neumann series. While NeumNet offers high-speed image reconstruction, the resulting images are still significantly impacted by blocking artifacts. AMP-Net incorporates the unfolding algorithm AMP into a neural network structure, extending its capabilities. TransCS, on the other hand, introduces a Transformer-based network built on ISTA that captures global dependencies between image sub-blocks while iteratively applying gradient descent and soft-thresholding operations. Furthermore, DRCAMP-Net [19] integrates AMP with extended residual convolution to mitigate block artifacts and broaden the receptive field.

Another approach focuses on deep learning models built on convolutional neural networks (CNNs). These models reconstruct images by stacking convolutional layers, prioritizing the retention of local image features. For example, DR2-Net [20] leverages linear mapping and residual networks for initial and final image reconstruction, while ReconNet achieves this directly through convolutional layers. DPA-Net [21] enhances reconstruction quality by preserving texture details, and MSCRLNet [22] uses multi-scale residual networks to improve shallow feature extraction by concentrating on channels. However, due to the inherent locality of convolutional layers, CNN-based models have limitations in capturing global positional relationships. To address global dependencies, these models often resort to inefficient stacking of convolutional layers to expand the receptive field. Thus, there is a clear need to establish a new DL-based image CS paradigm that effectively captures global relationships among image subblocks.

## 3. FusionOpt-Net Module

We propose a novel algorithmic framework that integrates the iterative process of FISTA with the feature extraction process of Transformer networks. This architecture combines the iterative algorithm of FISTA-Net with the deep self-attention mechanism of TransCS, achieving superior image reconstruction performance through technical fusion. The data flow is illustrated in Figure 1.

The core of the FusionOpt-Net model is the FISTA-based Transformer backbone. We customize the traditional FISTA by embedding it into the Transformer architecture. This customization allows the Transformer to effectively model the global dependencies among image subblocks, which are crucial for accurately reconstructing images from compressed measurements. In each iteration, the model performs a gradient descent update followed by a soft thresholding operation, which is a typical step in FISTA. This process is then integrated into the Transformer’s multi-head self-attention mechanism. By doing so, the model not only captures local image features but also effectively models the long-range dependencies across the entire image, which is essential for high-quality reconstruction.

In the model architecture diagram, from “stage 1” to “stage n”, each stage has a clear momentum update mechanism designed to accelerate the convergence of the model. As seen in the diagram, the output of each stage (after processing by the proximal mapping module) is combined with the output from the previous stage, and through a weighted summation (including the momentum term $\rho^{\left(k\right)}$), the input for the next stage is formed. This structure, by adding a momentum term, optimizes the current update step by utilizing a linear combination of the previous two iteration results during each iteration. This not only helps to accelerate convergence but also effectively mitigates oscillations during the iterative process. Our momentum module is designed as a learnable parameter, and during the entire network training process, these momentum parameters dynamically adjust according to the specific task requirements to ensure faster convergence and higher performance.

### 3.1. Sampling Module

In order to achieve better image reconstruction results, the sampling module of FusionOpt-Net utilizes a data-driven trainable sensing matrix. The sampling module uses a partition function $F_{B}\left(\cdot \right)$ to divide the original image x into B×B non-overlapping blocks, followed by a flattening function $F_{\mathrm{vec}}\left(\cdot \right)$ that projects the blocks into vectors. The sensing matrix φ is trained through backpropagation using training images, ultimately conforming to a Gaussian distribution. Therefore, the sampling module can be expressed as
(12)$$y=S\left(x,\varphi \right)=\varphi \cdot F_{\mathrm{vec}}\left(F_{B}\left(\cdot \right)\right)$$
where S(·,φ) signifies the sampling process. Compared to random sensing matrices, the learned ones are more efficient for hardware implementation and demand less storage capacity.

### 3.2. Reconstruction Module

The FusionOpt-Net reconstruction module includes two submodules: initial reconstruction and deep reconstruction.

#### 3.2.1. Initial Reconstruction

The initial reconstruction module is a key component of the FusionOpt-Net framework, with its primary task being the initial reconstruction of the image after sampling. This module is implemented through a trainable initial reconstruction matrix $\tilde{\varphi }$. The matrix $\tilde{\varphi }$ is initialized as the transpose of the sampling matrix φ, $\tilde{\varphi }=\varphi^{T}$. This initialization method leverages the structural information of the sampling matrix, contributing to the stability of the initial reconstruction. The sampled image representation y undergoes a linear transformation using the initial reconstruction matrix $\tilde{\varphi }$, yielding the initial reconstructed image $x_{\mathrm{int}}$. This process is expressed as
(13)$$x_{\mathrm{int}}=I\left(y,\tilde{\varphi }\right)=\tilde{\varphi }\cdot y$$
where $I(\cdot ,\tilde{\varphi })$ represents the initial reconstruction operation. Relying solely on the initial reconstruction module may lead to artifacts and missing details in the initial reconstructed image because the initial reconstruction process only performs a simple linear transformation. To improve reconstruction quality and reduce artifacts, the deep reconstruction module further refines the reconstruction based on this initial output.

#### 3.2.2. Deep Reconstruction

The deep reconstruction module D(·) is implemented using a Transformer backbone network and CNN based on FISTA. The Transformer backbone network guides the solution of the general $\ell_{1}$-norm optimization problem at each layer, where the threshold and shrinkage values are updated in each iteration. The momentum ρ is learned automatically from the training data. $r^{k}$ represents the residual in FISTA, while the current estimate $x^{k}$ is obtained from the previous estimate $x^{k-1}$.

Inspired by TransCS, we designed a function $F_{B}^{\prime }\left(\cdot \right)$ that partitions the B×B input into non-overlapping $B^{\prime }\times B^{\prime }$ blocks. The iterative shrinkage-thresholding operation is then expressed as
(14)$$x_{o}^{k}=F_{B^{\prime }}\left(r^{k}-\lambda^{k}\varphi^{T}\left(\varphi \cdot r^{k}-y\right)\right)$$
where $r^{k}$ is the input at the *k*-th iteration, $x_{c}^{k}$ denotes the output at the same iteration, and $\lambda^{k}$ is the step size updated at each iteration according to traditional FISTA.

Next, the pre-processing module (piling residual layers) refines the output $x_{c}^{k}$ to reduce noise and preserve high-quality details through convolutional layers, learned from training data. This process is expressed as
(15)$$x_{\mathrm{pre}}^{k}=F_{\mathrm{vec}}^{-1}\left(x_{0}^{k}\right)-C_{\mathrm{pre}}^{k}\left(F_{\mathrm{vec}}^{-1}\left(x_{0}^{k}\right)\right)$$
where $F_{\mathrm{vec}}^{-1}\left(\cdot \right)$ denotes the inverse vectorization function, and $C_{\mathrm{pre}}^{k}\left(\cdot \right)$ represents the *k*-th convolutional layer. The pre-processing module consists of six layers, each with a 3×3 kernel size. The first and last layers have one channel, while the middle FusionOpt-Net layers have 32 channels.

In the coding module of the Transformer, FusionOpt-Net deep reconstruction uses the embedded positions of image patches to compress the sequence of input tokens. The positional encoding (PE) is used to retain the spatial relationships between image patches. The final result is a matrix representing the encoded sequence, expressed as
(16)$$x_{\mathrm{en}}^{k}=T_{\mathrm{en}}^{k}\left(F_{p}\left(x_{\mathrm{pre}}^{k}\right)+PE\left(F_{p}\left(x_{\mathrm{pre}}^{k}\right)\right)\right)$$
where $T_{\mathrm{en}}^{k}$ is the Transformer encoder function, and $F_{p}\left(\cdot \right)$ represents a function that partitions the input into non-overlapping blocks. After encoding, the representation $x_{\mathrm{en}}^{k}$ undergoes element-wise soft thresholding to reduce noise and improve sparsity. This process is expressed as
(17)$$x_{\mathrm{soft}}^{k}=F_{\mathrm{sgn}}\left(x_{\mathrm{en}}^{k}\right)\cdot F_{\mathrm{act}}\left(x_{\mathrm{abs}}\left(x_{\mathrm{en}}^{k}\right)-\zeta^{k}\right)$$
where $F_{\mathrm{sgn}}\left(\cdot \right)$ is the sign function, $F_{\mathrm{act}}\left(\cdot \right)$ is an activation function, $F_{\mathrm{abs}}\left(\cdot \right)$ is the absolute value function, and $\zeta^{k}$ is the current threshold.

The result of the soft thresholding $x_{\mathrm{soft}}^{k}$ is combined with the pre-processed result $x_{\mathrm{pre}}^{k}$, and a weighted update is performed:(18)$$x_{\mathrm{de}}^{k}=x_{\mathrm{pre}}^{k}-\eta^{k}\cdot T_{\mathrm{de}}^{k}\left(x_{\mathrm{pre}}^{k},x_{\mathrm{soft}}^{k}\right)$$
where $\eta^{k}$ is the weight factor, and $T_{\mathrm{de}}^{k}\left(\cdot \right)$ represents the decoder function at the *k*-th iteration.

After the deep module, a post-processing module is designed, which is expressed as
(19)$$x_{\mathrm{post}}^{k}=F_{B^{\prime }}^{-1}\left(x_{\mathrm{de}}^{k}\right)-C_{\mathrm{post}}^{k}\left(F_{B^{\prime }}^{-1}\left(x_{\mathrm{de}}^{k}\right)\right)$$
where $F_{B}^{-1}\left(\cdot \right)$ represents the inverse partition function, and $C_{\mathrm{post}}^{k}\left(\cdot \right)$ denotes the convolutional layer configuration in the post-processing block.

The updated vectorization function $F_{\mathrm{vec}}\left(\cdot \right)$ reprojects the reconstruction result, allowing the image blocks to proceed to the next iteration:(20)$$x^{k+1}=F_{\mathrm{vec}}\left(x_{\mathrm{post}}^{k}\right)$$

Intermediate variables $r^{k+1}$ and $\rho^{k+1}$ are updated for the next iteration with momentum strategies to accelerate convergence:(21)$$r^{k+1}=x^{k+1}+\rho^{k}\left(x^{k+1}-x^{k}\right)$$

The final reconstruction result is obtained by applying the inverse vectorization function $F_{B}^{-1}\left(\cdot \right)$ to the final iteration output:(22)$$\hat{x}=F_{B}^{-1}\left(F_{\mathrm{vec}}^{-1}\left(x^{n+1}\right)\right)$$

The parameter changes during each iteration can follow a predetermined pattern. Consequently, we present Algorithm 1 to illustrate the reconstruction process.
**Algorithm 1** Forward Propagation for Image Recover**Require:** number of iteration stages *n*, initial reconstruction matrix $\tilde{\varphi }$, soft thresholds $\zeta^{1∼n}$,   weight coefficients $\eta^{1∼n}$, iteration step size $l^{1∼n}$, measurements y, scalar for momentum   update $\rho^{1∼n}$, sensing matrix φ
**Ensure:** reconstructed image $\hat{x}$
  1:**Trainable hyperparameters**: $\tilde{\varphi },l^{1∼n},t^{1∼n},\eta^{1∼n}$  2:**Initialization**: $X_{\mathrm{init}}=I\left(y,\tilde{\varphi }\right)$  3:**Begin the iteration**: $k=1,r^{1}=X_{\mathrm{init}}=x^{1}$  4:**while** k≤n **do**  5:    $x_{o}^{k}=F_{B^{\prime }}\left(r^{k}-\lambda^{k}\varphi^{T}\left(\varphi \cdot r^{k}-y\right)\right)$  6:    $x_{\mathrm{pre}}^{k}=F_{\mathrm{vec}}^{-1}\left(x_{o}^{k}\right)-C_{\mathrm{pre}}^{k}\left(F_{\mathrm{vec}}^{-1}\left(x_{o}^{k}\right)\right)$  7:    $x_{\mathrm{en}}^{k}=T_{\mathrm{en}}^{k}\left(F_{p}\left(x_{\mathrm{pre}}^{k}\right)+PE\left(F_{p}\left(x_{\mathrm{pre}}^{k}\right)\right)\right)$  8:    $x_{\mathrm{soft}}^{k}=F_{\mathrm{sgn}}\left(x_{\mathrm{en}}^{k}\right)\cdot F_{\mathrm{act}}\left(x_{\mathrm{abs}}\left(x_{\mathrm{en}}^{k}\right)-\zeta^{k}\right)$  9:    $x_{\mathrm{de}}^{k}=x_{\mathrm{pre}}^{k}-\eta^{k}\cdot T_{\mathrm{de}}^{k}\left(x_{\mathrm{pre}}^{k},x_{\mathrm{soft}}^{k}\right)$10:    $x_{\mathrm{post}}^{k}=F_{B^{\prime }}^{-1}\left(x_{de}^{k}\right)-C_{post}^{k}\left(F_{B^{\prime }}^{-1}\left(x_{de}^{k}\right)\right)$11:    $x^{k+1}=F_{\mathrm{vec}}\left(x_{\mathrm{post}}^{k}\right)$12:    $r^{k+1}=x^{k+1}+\rho^{k}\left(x^{k+1}-x^{k}\right)$13:    k=k+114:**end while**15:$\hat{x}=F_{B}^{-1}\left(F_{\mathrm{vec}}^{-1}\left(x^{n+1}\right)\right)$


### 3.3. Loss Function

During the training of FusionOpt-Net, we simultaneously refine the sampling module S(·,φ) and the recover module $D(I\left(\cdot ,\tilde{\varphi }\right))$, with the original images serving as both inputs and training labels. The parameters to be trained for the *k*-th stage of the deep reconstruction are denoted by $\omega^{k}$, while for the *n* stages, the collective trainable parameters are indicated by $\omega^{1∼n}$. To automatically train the initialization and deep reconstruction modules from the measured values y, we measure the differences between the source and recovered images using mean squared error (MSE). We define the loss function as
(23)$$L_{\mathrm{total}}\left(\varphi ,\tilde{\varphi },\omega^{1∼n}\right):=\frac{1}{2n}\sum_{i=1}^{n}{∥D\left(I\left(S\left(x_{i},\varphi \right),\tilde{\varphi }\right)\right)-x_{i}∥}_{2}^{2}$$
where $x_{i}$ represents the *i*-th training image, and *n* is the total number of training images.

## 4. Experimental Results

In this section, several experiments are conducted to verify the performance of the proposed method. Firstly, in Section 3.2, a comparison between the FusionOpt-Net method and other models is performed on several public datasets. Subsequently, in Section 3.3, the robustness of the FusionOpt-Net method is tested on multi-level Gaussian noise images.

### 4.1. Experimental Settings

The FusionOpt-Net training dataset is derived from the BSD500 dataset [23], comprising 200 training images, 100 validation images, and 200 testing images. The validation dataset we use is Set11. We randomly segment images in the training dataset into 200 sub-images, each measuring 96 × 96 pixels, creating a total of 100,000 sub-images. To augment the data, we apply random horizontal and vertical flips, rotations, and scaling to enhance image diversity. The experimental results are evaluated using three widely used benchmarks: Set11 [24], BSD200 [23], and Urban100 [25].

The FusionOpt-Net training process follows the same settings as the DL-based CS method (such as ISTA-Net). The patch size *P* is set to 8, the initial step size is 1.0, and the regularization parameter λ is initialized to 0.1. The initial value of ρ is 0.01, and the number of iterations *H* is set to 8. Training is conducted for 200 epochs with a batch configuration of 64. The learning rate is set to decay from the 101st to the 150th epoch, and the last 50 epochs are trained with a constant learning rate. We use the Adam optimizer for training. The FusionOpt-Net model is compared with several state-of-the-art methods, including CSformer [26], ISTA-Net+ [11], CSNet [27], AMP-Net [28], and TransCS, which are all based on traditional algorithms combined with deep learning models. Performance evaluation is carried out using perceptual metrics, PSNR, and SSIM. The better performance of the method is indicated by higher PSNR and SSIM values. The models compared to FusionOpt-Net are obtained from their respective sources and executed with default configurations. To ensure an unbiased evaluation, all training images for the rival models are sourced from the BSD500 dataset. The experiments are conducted using the PyTorch 1.9.0 framework on a system with an Intel Xeon 8336 CPU and a GeForce RTX 4090 GPU.

### 4.2. Comparisons with State-of-the-Art Methods

In our study, we conducted a comprehensive evaluation of Csformer, ISTA-Net+, AMP-Net, CsNet, TransCS, and our proposed model across the Set11, BSD200, and Urban100 datasets at sampling rates of 0.04, 0.1, 0.25, and 0.5. The evaluation metrics used were peak signal-to-noise ratio, (PSNR, dB) and structural similarity index (SSIM). Table 1 presents the detailed experimental results.

The FusionOpt-Net model consistently demonstrated significant advantages across all datasets and sampling rates:Set11 Dataset: At a sampling rate of 0.04, our model achieved a PSNR of 25.34 and SSIM of 0.7815, both superior to other models. At higher rates like 0.5, our model further demonstrated superiority with a PSNR of 39.91 and SSIM of 0.9809, notably higher than ISTA-Net+ (38.07) and TransCS (38.88).BSD200 Dataset: Across various sampling rates, our model consistently outperformed others. For instance, at a rate of 0.25, our model achieved a PSNR of 31.91 and SSIM of 0.9237, surpassing TransCS (PSNR 31, SSIM 0.9171) and ISTA-Net+ (PSNR 29.51, SSIM 0.8659). At a 0.5 sampling rate, our model reached a PSNR of 37.06 and SSIM of 0.9748, reaffirming its superior performance.Urban100 Dataset: At a low sampling rate of 0.04, our model led with a PSNR of 22.05 and SSIM of 0.6619. At a 0.5 sampling rate, our model achieved a PSNR of 35.51 and SSIM of 0.9758, significantly surpassing ISTA-Net+ (PSNR 34.58, SSIM 0.9661) and TransCS (PSNR 34.16, SSIM 0.9687).

On average, our model exhibited the highest PSNR (32.32) and SSIM (0.9), significantly outperforming other models. These results underscore the capability of our model to consistently deliver high-quality reconstructed images across different datasets and sampling rates. Moreover, its robust performance at low sampling rates (e.g., 0.04 and 0.1) highlights its efficacy in sparse data scenarios.

Visual comparisons between our model and competing methods further validate our findings. As shown in Figure 2, Our approach excelled in detail preservation, texture reconstruction, and edge sharpness, notably outperforming ISTA-Net+, AMP-Net, and TransCS. Specifically, our model accurately reproduced complex structures, such as natural shadow transitions in portrait images and sharp patterns in butterfly wings, reducing blurring effects significantly compared to other methods.

In conclusion, our model exhibits superior performance in compressive sensing image reconstruction tasks, as evidenced by both quantitative metrics (PSNR, SSIM) and qualitative visual assessments. These findings underscore the effectiveness and potential application value of our proposed method in the field of image reconstruction.

### 4.3. Noise Robustness

To assess image reconstruction robustness in various noisy environments, Gaussian noise with a mean of zero and standard deviations σ∈0.001,0.002,0.004 was added to the BSD200 test dataset. The noise robustness of FusionOpt-Net was compared with three deep learning models (ISTA-Net+, AMP-Net, and TransCS). PSNR and SSIM metrics were used for evaluation at four sampling rates ϵ∈0.04,0.1,0.25,0.5, along with visual noise level comparisons. Additionally, average PSNR and SSIM values for different noise levels are provided for all four reconstruction methods.The results are shown in Table 2.

Firstly, under different noise levels, the average peak signal-to-noise ratio (PSNR) and structural similarity index (SSIM) are compared. Regardless of the noise level, the FusionOpt-Net model achieves the highest PSNR and SSIM values in most situations. This indicates that the FusionOpt-Net model maintains superior image quality and structural details over other models in varying noise levels. For instance, at a noise level of σ=0.004, the PSNR of the FusionOpt-Net model only decreases by 2.86% from 28.26 to 23.68, while the PSNR of ISTA-Net+ dropped from 25.76 to 23.68. These data indicate that the FusionOpt-Net model maintains a higher PSNR in noisier environments and outperforms other models in preserving visual structures. The FusionOpt-Net model is able to effectively suppress noise and preserve the high-frequency information of the image structure under various noise conditions.

In particular, under high noise levels and sample rates, the FusionOpt-Net model still demonstrates excellent visual consistency and structural delineation.As shown in Figure 3. Compared to ISTA-Net+, AMP-Net, and TransCS, the FusionOpt-Net model shows stronger robustness and reliability in various noise environments, achieving superior visual effects and reliable results in practical applications.

### 4.4. Complexity Analysis

We conduct a model complexity analysis of MMR-CSNet and several competing methods (ISTA+, CSNet, CSformer, AMP-9BM, TransCS) across three dimensions: average runtime, the number of giga floating-point operations (GFLOPs), and the number of parameters. The average runtime assesses the time required for the model to compress and reconstruct an image. GFLOPs are used to evaluate the computational complexity, while the number of parameters reflects the spatial complexity of the model. These metrics are derived by forward propagating a single 256 × 256 image at a 0.1 sampling rate. As illustrated in Table 3 and Figure 4.

FusionOpt-Net achieves a computational time of 0.026 s on the RTX 4090D GPU for τ=0.1, making it highly efficient and suitable for real-time applications. Although slightly slower than the fastest model, CSNet (0.008 s), it remains competitive with methods like ISTA-Net+ (0.023 s) and AMP-Net (0.017 s), highlighting a balance between complexity and performance. The parameter count of 1.445 Mb, comparable to TransCS (1.489 Mb), reflects FusionOpt-Net’s enhanced feature extraction capabilities, justifying the trade-off for improved reconstruction quality and flexibility. With a moderate computational complexity of 12.011 GFLOPs, FusionOpt-Net is optimized for efficiency without compromising performance, making it a strong candidate for scenarios requiring high precision and resource-conscious deployments.

### 4.5. Ablation Studies

To verify the efficacy of the measurement reuse strategy, we further conduct ablation studies on BSDS100. The models compared include FusionOpt-Net and FusionOpt-Net without the momentum module. From the results, as shown in Figure 5, we can observe the following: The momentum module is useful for improving the reconstruction quality of an image. It plays a more important role, especially at high sampling rates. This is probably because the module acts as a residual-like structure in the overall structure, which improves the stability of the deep learning model during image reconstruction, resulting in a higher quality of the recovered result, which is more similar to the original image.

## 5. Future Work

The performance of FusionOpt-Net is potentially restricted by the fixed sampling block size used in the Transformer backbone. This fixed size may limit the flexibility and adaptability of the model to different types of images or scenarios where variable block sizes could be more effective. While FusionOpt-Net shows improved robustness to noise compared to other models, there is still room for enhancing its performance in extremely noisy environments. This suggests that the model’s ability to handle varying levels of noise is not fully optimized.

Future research could focus on developing a Transformer-based CS method with a dynamic and adaptive block size. This strategy would allow the model to adjust the block size based on the sampling matrix and the specific characteristics of different image areas, potentially leading to improved reconstruction performance. Another area of future work involves exploring more robust modifications of the Transformer architecture specifically designed for noisy image reconstruction scenarios. Enhancing the model’s ability to maintain high reconstruction quality in the presence of significant noise could further expand its applicability in real-world situations.

## 6. Conclusions

This paper introduces a novel compressed sensing image reconstruction algorithm that integrates the FISTA and Transformer networks. By combining the fast convergence properties of FISTA with the powerful feature extraction capabilities of Transformer networks, we have developed an efficient and high-quality image reconstruction method. The experimental results demonstrate that the FusionOpt-Net model exhibits significantly superior reconstruction performance across multiple image datasets, outperforming existing models such as ISTA-Net+ and TransCS in terms of metrics like PSNR and SSIM. Particularly noteworthy is its ability to preserve fine details and suppress noise effectively, especially in scenarios with high noise levels and low sampling rates, showcasing robustness in diverse environments.

In comparison to traditional algorithms, the FusionOpt-Net model not only addresses the complexity of hyperparameter tuning but also leverages deep learning to automatically learn features from data, thereby substantially enhancing image reconstruction quality. The future work will focus on further optimizing algorithmic efficiency and exploring its potential in other compressed sensing applications, aiming to achieve efficient and high-quality image reconstruction in broader contexts. This study provides new insights into advancing compressed sensing reconstruction algorithms and establishes a solid foundation for practical image reconstruction tasks.

## Figures and Tables

**Figure 1 sensors-24-05976-f001:**
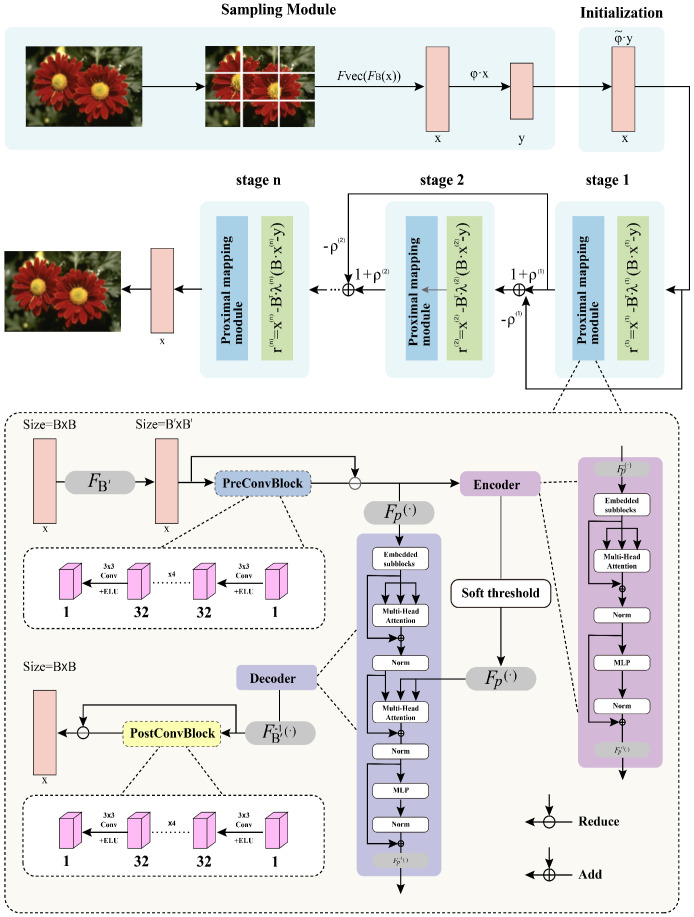
FusionOpt-Net framework.

**Figure 2 sensors-24-05976-f002:**
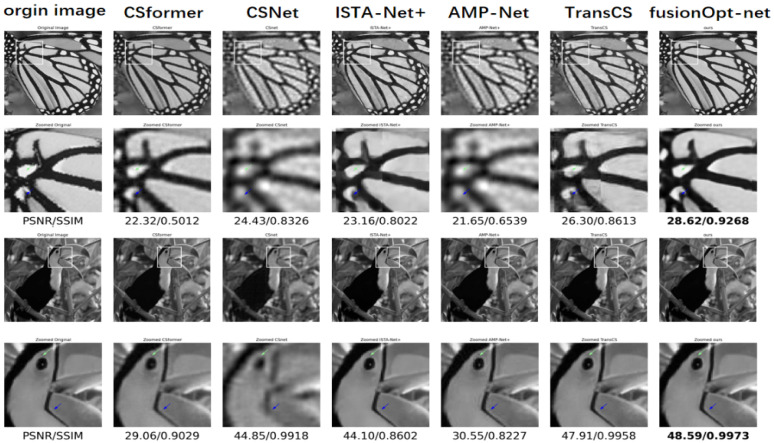
Reconstruction results for butterfly and bird images using FusionOpt-Net and other methods. Sampling rates τ are 0.04 for the first row and 0.25 for the second row. Please zoom in for better comparison.

**Figure 3 sensors-24-05976-f003:**
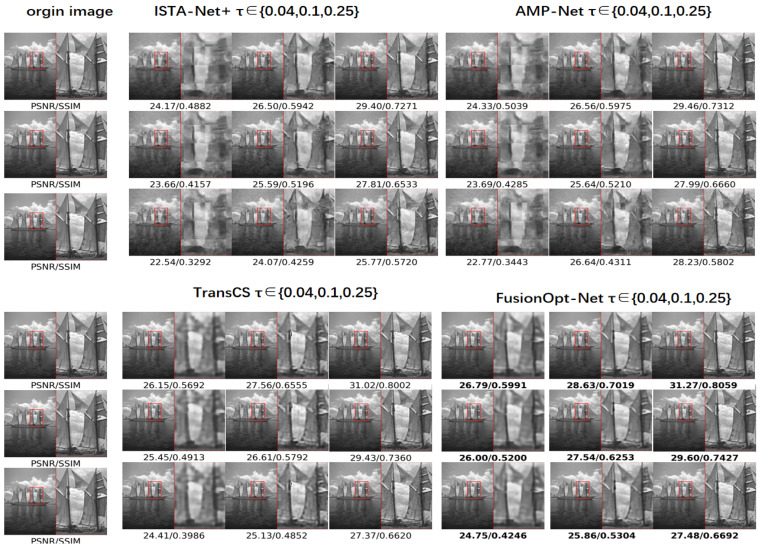
Noise Robustness Comparison. Visual analysis of different image CS methods on cactus and ship images from the BSD100 dataset at sampling rates τ∈0.04,0.10,0.25. Gaussian noise with variances σ∈0.001,0.002,0.004 was introduced. Note the effectiveness in recovering the ship images.

**Figure 4 sensors-24-05976-f004:**
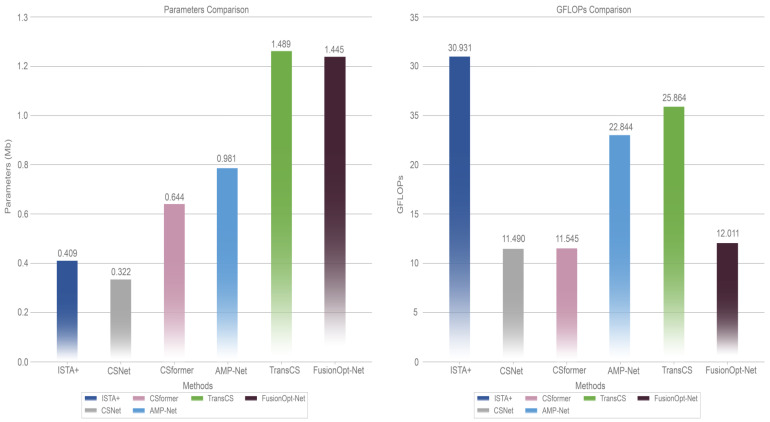
Comparison of the number of GFLOPs required to run a 256 × 256 pixel image in the model and the number of model parameters for τ=0.1.

**Figure 5 sensors-24-05976-f005:**
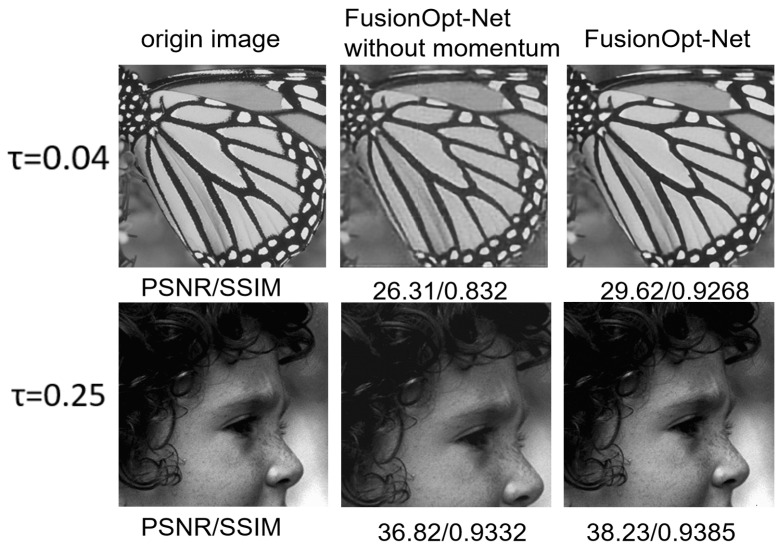
Comparison of visualizations with and without momentum at different sampling rates τ∈0.04,0.25.

**Table 1 sensors-24-05976-t001:** PSNR (dB) and SSIM assessment for various models on Set11, BSD200, and Urban100 datasets at different sampling rates τ∈0.04,0.1,0.25,0.3,0.4,0.5. Best performances are highlighted in bold.

Datasets	Models	0.04		0.1		0.25		0.5		Avg.	
		PSNR	SSIM	PSNR	SSIM	PSNR	SSIM	PSNR	SSIM	PSNR	SSIM
set11	Csformer	23.98	0.7388	26.57	0.8411	32.13	0.9204	38.92	0.9876	29.5	0.829
	ISTA-Net+	21.56	0.624	26.49	0.8036	32.44	0.9237	38.07	0.9706	29.64	0.8305
	AMP-Net	21.68	0.6223	25.92	0.7828	32.05	0.9139	37.77	0.9666	29.36	0.8214
	CsNet	22.34	0.6788	26.52	0.7983	32.78	0.9178	38.63	0.9752	30.07	0.8425
	TransCS	23.62	0.7253	26.64	0.8389	33.69	0.9444	38.88	0.9766	30.71	0.8713
	FusionOpt-Net	**25.34**	**0.7815**	**29.51**	**0.8869**	**34.51**	**0.9508**	**39.91**	**0.9809**	**32.32**	**0.9**
BSD200	Csformer	23.78	0.6577	25.9	0.7783	29.64	0.8947	35.21	0.9612	28.63	0.823
	ISTA-Net+	22.19	0.5682	25.21	0.7149	29.51	0.8659	34.57	0.9509	27.87	0.775
	AMP-Net	22.48	0.5836	25.26	0.7108	29.58	0.8591	34.8	0.9489	28.03	0.7756
	CsNet	23.22	0.6157	25.68	0.7457	30.02	0.9023	35.02	0.9578	28.49	0.8054
	TransCS	23.86	0.6634	26.04	0.7804	31	0.9171	35.83	0.9698	29.18	0.8327
	FusionOpt-Net	**25.1**	**0.7014**	**27.88**	**0.8253**	**31.91**	**0.9237**	**37.06**	**0.9748**	**30.49**	**0.8563**
Urban100	Csformer	20.22	0.5243	22.32	0.6788	28.2	0.8842	34.02	0.9597	26.19	0.7618
	ISTA-Net+	18.9	0.4913	22.6	0.699	28.26	0.8873	34.58	0.9661	26.09	0.7609
	AMP-Net	19.19	0.4918	22.22	0.668	27.68	0.8678	34.25	0.9606	25.84	0.7471
	CsNet	19.23	0.5022	22.46	0.6981	27.91	0.8834	34.43	0.9644	26.01	0.762
	TransCS	20.99	0.598	23.29	0.7483	29.26	0.9196	34.16	0.9687	26.93	0.8087
	FusionOpt-Net	**22.05**	**0.6619**	**25.49**	**0.8259**	**30.26**	**0.9308**	**35.51**	**0.9758**	**28.33**	**0.8486**

**Table 2 sensors-24-05976-t002:** PSNR (dB) and SSIM comparisons on BSD200 with different noise levels σ and various sample rates τ. Highlight the best performance in bold.

σ	τ	ISTA-Net+		AMP-Net		TransCS		FusionOpt-Net	
		PSNR	SSIM	PSNR	SSIM	PSNR	SSIM	PSNR	SSIM
0.001	0.04	21.63	0.4591	21.92	0.4688	23.68	0.556	**24.36**	**0.5849**
	0.1	24.13	0.59	24.21	0.5852	25.45	0.6667	**26.55**	**0.7086**
	0.25	27.24	0.7407	27.33	0.7343	29.17	0.8175	**29.38**	**0.822**
	0.5	30.63	0.8646	30.75	0.8571	32.4	0.9005	**32.74**	**0.9067**
	Avg.	25.9	0.6636	26.05	0.6614	26.68	0.7352	**28.26**	**0.7556**
0.002	0.04	21.22	0.4045	21.52	0.4118	23.22	0.4969	**23.83**	**0.5244**
	0.1	23.45	0.5295	23.55	0.5248	24.76	0.6065	**25.73**	**0.6486**
	0.25	26.1	0.6827	26.22	0.6773	27.99	0.7682	**28.14**	**0.7728**
	0.5	29.09	0.8262	29.24	0.8194	30.84	0.8697	**31.1**	**0.8764**
	Avg.	24.96	0.6107	25.13	0.6082	26.71	0.6853	**27.2**	**0.7056**
0.004	0.04	20.53	0.338	20.85	0.3428	22.46	0.423	**22.97**	**0.4486**
	0.1	22.43	0.4567	22.56	0.4522	23.69	0.5314	**24.52**	**0.5732**
	0.25	24.56	0.6144	24.76	0.612	26.39	0.7069	**26.48**	**0.7122**
	0.5	27.21	0.7829	27.39	0.7774	28.86	0.8328	**29.06**	**0.84**
	Avg.	23.68	0.548	23.89	0.5461	25.35	0.6235	**25.76**	**0.6435**

**Table 3 sensors-24-05976-t003:** Average running time comparison of different methods on RTX 4090D with τ=0.1.

Methods	τ=0.1 GPU	Platform
ISTA-Net+	0.023	**GPU: RTX 4090D**
CSNet	0.008	
CSformer	0.046	
AMP-Net	0.017	
TransCS	0.027	
FusionOpt-Net	0.026	

## Data Availability

The data used to support the findings of this study are available from the corresponding author upon request.

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
