# Peer review of "FusionOpt-Net: A Transformer-Based Compressive Sensing Reconstruction Algorithm"

_sensors, 2024, doi:10.3390/s24185976_

Round 1

Reviewer 1 Report

Comments and Suggestions for Authors

The paper introduces a novel image reconstruction algorithm that combines the Fast Iterative Shrinkage-Thresholding Algorithm (FISTA) with Transformer networks for Compressive Sensing (CS). The authors propose a block-based sampling approach and map FISTA's iterative process onto neural networks to improve reconstruction efficiency and address hyperparameter challenges. The Transformer network's feature extraction capabilities are leveraged to enhance image reconstruction quality. However, the paper still has the following problems:

Could the authors elaborate on the implementation details of the FusionOpt-Net model, particularly the integration of FISTA with Transformer networks?

Are there any specific ablation studies planned to understand the contribution of each component of the FusionOpt-Net model?

How does the computational complexity of the proposed method compare to existing CS reconstruction methods, and how does it scale with data size?

The sentence in the introduction, "This necessitates the development of efficient solutions that can support IoT applications." lacks context and needs to cite the latest relevant papers, such as "Image privacy protection scheme based on high-quality reconstruction DCT compression and nonlinear dynamics"

Has the model been tested on other types of data or in different application scenarios to evaluate its generalization capability?

Can the authors discuss potential practical applications of the FusionOpt-Net model in real-world environments?

Is there any theoretical analysis that supports the proposed method's advantages over traditional CS reconstruction algorithms?

Are there any limitations to the proposed method that the authors are aware of, and what are the potential areas for future research?

Reviewer 2 Report

Comments and Suggestions for Authors

While the paper presents a commendable effort in proposing a novel Compressive Sensing image reconstruction algorithm that combines the Fast Iterative Shrinkage-Thresholding Algorithm (FISTA) with Transformer networks, several areas require major improvement:  

1) Articulation of Challenges and Motivation: The paper lacks a clear and specific articulation of the challenges and the motivation behind the proposed approach. While the combination of keywords such as "Transformer," "compressive sensing," and "reconstruction" is relatively rare, with only one notable example ([a] Dual-Path Image Reconstruction: Bridging Vision Transformer and Perceptual Compressive Sensing Networks, 2023), the discussion of related challenges feels overly broad and could easily encompass the methods being compared. This lack of specificity diminishes the distinct motivation for the proposed approach.  

2) Insufficient Literature Review: The paper, as a full-length manuscript, lacks a dedicated section on related work, which is inappropriate given the scope of the research. Even with the references included in the introduction, there are only around ten citations, with a total of just seventeen references. This level of literature review is insufficient to draw solid and well-supported conclusions.  

3) Typos and Formatting Issues: There are several typographical and formatting issues throughout the manuscript, such as the misalignment of "where" in line 167. These structural and formatting problems detract from the overall readability and professionalism of the paper.  

4) Selection of Hyperparameters: The selection of hyperparameters in Table 2, particularly for sigma (σ) and tau (τ), appears arbitrary. The values chosen are neither evenly spaced nor follow a geometric progression, nor do they cover the typical range of parameter selections in the field. Additionally, there is no indication that these choices follow an established methodology. This gives the impression of a lack of rigor in the experimental setup.  

5) Lack of Qualitative Analysis and Figure References: Many figures and tables are not adequately referenced in the text, particularly Figures 2 and 3, which also lack qualitative analysis. For example, the visual similarities between ISTA-Net+ and the proposed method in Figure 2 are not sufficiently explained. The authors fail to guide the reviewer and the reader through the rationale behind these comparisons, leaving it up to the reader to infer the significance—an approach that, from my perspective, is not effective.  

Overall, while the work is interesting, it would require substantial revisions to meet the standards expected of this journal. The experimental work is currently somewhat weak, and the paper's organization could be significantly improved.

Round 2

Reviewer 1 Report

Comments and Suggestions for Authors

The author responded to the reviewer's questions point-to-point, and I have no further comments.